# Evaluating the Impact of the Nirvana Fitness and Functional Training Programs on Young Women’s State Body Appreciation and Its Correlates

**DOI:** 10.3390/healthcare12161632

**Published:** 2024-08-16

**Authors:** Rasa Jankauskiene, Vaiva Balciuniene, Renata Rutkauskaite, Simona Pajaujiene, Migle Baceviciene

**Affiliations:** 1Institute of Sport Science and Innovations, Lithuanian Sports University, 44221 Kaunas, Lithuania; 2Department of Physical and Social Education, Lithuanian Sports University, 44221 Kaunas, Lithuania; vaiva.balciuniene@lsu.lt (V.B.); renata.rutkauskaite@lsu.lt (R.R.); migle.baceviciene@lsu.lt (M.B.); 3Department of Coaching Science, Lithuanian Sports University, 44221 Kaunas, Lithuania; simona.pajaujiene@lsu.lt

**Keywords:** state mindfulness in physical activity, embodiment, positive body image, self-objectification, self-determination

## Abstract

This study examined the effects of the Nirvana Fitness (NF) program on state body appreciation (BA) and its correlates, as well as the mechanisms that could explain these changes. The study participants were allocated to NF (n = 21) and functional training (control, n = 22) groups and participated in an 8-week exercise intervention. The mean age was 24.4 ± 6.7. The participants filled out online questionnaires on state measures of BA, body surveillance (BS), functionality appreciation, body–mind connection, mindfulness in physical activity (SMS-PA), intrinsic exercise motivation (IM), satisfaction of basic psychological needs, and perceived physical fitness (PPF) immediately after the first and last sessions. The results revealed a significant improvement in state BA and its correlates in both groups, with no changes in PPF in the NF group or IM in either group. An interaction between group and time effects was revealed in a change of relatedness, indicating a stronger effect in the NF group. Positive changes in SMS-PA, BS, and satisfaction of the need for relatedness significantly predicted improvement in state BA. These findings highlight the importance of targeting mindfulness, body surveillance, and relatedness during physical activity in interventions aimed at promoting positive body image in university-aged women.

## 1. Introduction

### 1.1. The Effect of Physical Activity on the Body Image of Women and Girls

Insufficient physical activity in women and girls is more common than in men and boys [1,2]. Body dissatisfaction, self-objectification, teasing about appearance, physique anxiety, and the internalization of stereotyped body ideals are among the main reasons why women and girls do not participate in physical activity [3,4,5,6]. On the other hand, positive body image is related to higher physical activity in women and girls [7,8,9]. Positive body image is a multidimensional construct that includes acceptance, appreciation, and respect for one’s own body, inner positivity, and a broad conceptualization of human beauty [10]. Positive body image is related to adaptive well-being constructs (self-esteem, self-compassion, and sexual satisfaction), general psychological well-being, and health-related behaviors in women and girls [11,12,13]. Physical activity and exercise are effective strategies for improving body image [14]. Previous meta-analyses have reported that exercise positively affects body image, with improvements of small to moderate effect size [15,16,17,18].

The possible mechanisms through which physical activity might positively affect the body image of women and girls are related to increased perceived physical fitness, body appearance satisfaction, self-esteem, body–mind connection, and decreased body surveillance [9,14,19,20,21,22]. However, a recent systematic review that brought together all the results on women and girls concluded that the mechanisms for the potential or actual changes in body image are not explained sufficiently in the literature and that it is not always clear whether changes in body image are related to changes in the perceptual, cognitive, affective, or behavioral domains [17]. Thus, it is important to provide more empirical data on this issue.

Finally, while, generally, exercise has been proposed as a potential strategy for increasing positive body image and decreasing self-objectification (the tendency to view yourself as a physical object, to monitor your own appearance from the third person’s perspective, and to value appearance over body functionality), some evidence exists that specific characteristics of exercise experience (mirrored walls, appearance-related comments by fitness coaches, and sports apparel) can result in increased negative body image [23,24,25]. Thus, the improvements in body image might depend on the degree to which exercise participants invest mental effort to conform to sociocultural expectations about appearance versus situations in sports environments where attention is fully focused on the movement itself. Thus, it is important to assess the impact of different exercise types on situational body image and related variables, including mindfulness, during physical activity.

### 1.2. Embodiment and Mindfulness during Physical Activity and Their Effect on Positive Body Image

The Developmental Theory of Embodiment (DTE) [26] states that embodiment is a form of experiencing and understanding the world through the positive and negative lived experiences of the body. In positive embodiment, the states of the inner physiological and psychological needs are attuned in an inhabiting way that provides a close, connected relationship with the body; through this, one can effectively know, respect, and voice body experiences and needs and express social power [27]. Positive embodiment prevents a person from looking at himself externally or objectifying himself. The fundamental element of positive embodiment is a deep and authentic body–mind connection and the ability to hear and prioritize the voice of the body over external requirements and social demands [28]. Previous results have shown that a good connection between body–mind is associated in women with higher body awareness, body responsiveness, and body appreciation; lower body surveillance; higher self-esteem; lower disordered eating; a lower body mass index; and higher physical activity [29].

Embodiment as the interaction between the mind, the body, and the external world reflects the Buddhist conceptualization of mindfulness [30]. In western understanding, mindfulness is conceptualized as the process of paying attention to the present moment and doing this with acceptance and without making judgments [31]. According to the Monitor and Acceptance Theory (MAT) [32], attention-monitoring skills, together with acceptance skills, enhance awareness of the experience of the present moment and reduce negative reactions, including anxiety, depression, and stress-related outcomes.

Mindfulness might be conceptualized as a difference between individuals (a trait or general mindfulness) or as a momentary experience (state mindfulness) that can vary between people in different contexts and times [30,33]. Global mindfulness (a trait) represents a general disposition towards mindfulness across varied contexts and moments in daily life [33]. Higher states of mindfulness may, over time, become manifest as an increased trait of mindfulness. Thus, state mindfulness might be the modifiable target of interventions when people are trained to pay attention to the present moment in a particular way.

People might possess different states of mindfulness when participating in physical activity, and different physical activities might differ in their promotion of mindfulness in exercisers [34]. For example, yoga participants report higher mindfulness during physical activity than do participants in other types of exercise [35]. Higher mindfulness in physical activity is related to enhanced positive body image. There is evidence that mindfulness during physical activity is associated with higher state positive body image (body appreciation), internal exercise motivation, and greater involvement in physical activity for health reasons than for appearance reasons [35,36,37]. There is initial evidence that mindfulness during physical activity mediates the association between participation in physical activity and positive body image [38]. Mindfulness during physical activity lets exercisers develop higher self-control and self-confidence through the increased non-judgmental acceptance of negative physical feelings (such as discomfort during physical activity) and the mental acceptance of negative thoughts related to body and movement (shame, fatigue, pain, and exertion) [39]. Non-judgmental monitoring and acceptance of mind and body is extremely important for novice exercisers, for people whose body weight is higher than the healthy range, and for those with a body disfigurement that does not meet the sociocultural expectations for appearance.

Some physical activities are considered to be more embodying and mindful than others [34]. Mindful physical activities are focused on the processes of becoming more internally connected, internally motivated, and focused on inner thoughts and the feelings of the body instead of external things such as body appearance [40]. Recent studies have concluded that yoga might be considered an embodying and mindful activity [19,40,41,42,43,44,45]. Evidence exists that participation in yoga increases general mindfulness [46] and that experienced yoga participants have higher general mindfulness than those with no experience [47] and than beginners [48]. Experimental studies have shown that participation in yoga has a positive impact on body–mind connectedness and body appreciation [42]. There is initial evidence that yoga affects positive body image through increased body–mind connection and lower body surveillance in adult women and adolescents [19,49].

In modern society, a variety of yoga-based practices have been cultivated [50]. These practices can be ancient, classical (Yoga, Tai Chi, Qigong), contemporary, or modern (Pilates, Callanetics), but their main distinguishing feature is that they involve physical activity of low-to-moderate intensity that relies on self-monitoring and a profound inward mental non-judgemental focus and is performed with a meditative, proprioceptive, or sensory-awareness component. Nirvana Fitness (NF) is a modern yoga-based practice that involves a combination of movement sequences, conscious regulation of the breath, and various techniques to improve attentional focus. It leads its participants to a slower and deeper daily breathing pattern (diaphragmatic) that should replace their shallow “default” breathing. NF promotes pursed-lip breathing (PLB) with prolonged exhalation. PLB allows people to control their oxygenation and ventilation [51]. This technique creates back pressure, producing a small amount of positive end-expiratory pressure and better gas exchange [52].

It is common for meditation to be practiced in a stable position (sitting or lying down), but NF brings meditation into the movement flow. By comparison, yoga emphasizes flexibility, strength, and mindfulness through various poses and breathing techniques. Both practices promote relaxation, but NF may have a stronger emphasis on rhythmic movement and the integration of music, while yoga often includes static poses and meditation without music. There is currently significant interest in the fields of music, health, and well-being. Music interventions, as a form of music therapy, can provide positive psychological and/or physiological benefits [53]. It is thought that listening to music is a means of stress reduction [54], inhibits the sympathetic nervous system, increases the release of endorphins, and decreases the circulating catecholamines [55]. Together with mindfulness-based exercise, music might support the enjoyment and motivation of healthy exercisers [56,57]. NF has developed unique sound technology with the help of sound researchers and engineers; it uses Theta (4–7 Hz) waves to induce a deep and effective relaxation state through direct influence on the listener’s brainwaves. NF soundscapes include soothing chillout-style music accompanied by breathing sound cues (using the sounds of Tibetan bowls) to indicate exactly how to breathe. Overall, the music adds an extra dimension to the NF experience, making it not just a physical workout but also a mental and emotional journey. This breathing-fitness-to-music concept, as an alternative fitness practice, is widely applicable to various groups of participants, especially novices and people with special needs or clinical conditions. Each exercise has a basic and a modified version, which gives instructors the possibility of adapting the exercises to less or more capable clients. NF has become increasingly popular globally because of its simplicity and orientation towards stress relief, internal body awareness, and improvements in strength and flexibility. However, despite its popularity, NF as a mindfulness-based activity has never been tested for its impact on changes in state body appreciation.

State mindfulness studies are not well-represented in the physical activity literature [33]. Most of the empirical studies in the physical activity domain focus on trait mindfulness and do not assess mindfulness as a state construct [33]. Thus, measuring the effects of NF on state body appreciation and its correlates is important since it expands our understanding of the types of physical activities and mechanisms through which embodying exercise activities promote positive body image. This might inform future intervention programs promoting positive body image and physical activity in women and girls.

### 1.3. The Effect of Mindfulness, Intrinsic Motivation, and Satisfaction during Physical Activity on Positive Body Image

Based on the Self-Determination Theory (SDT) [58], there is evidence that the relationship between physical activity and body image depends on the quality of motivation for physical activity [59]. Self-determined and autonomous exercise motivation supports the satisfaction of basic psychological needs (BPNs) such as autonomy, competence, and relatedness [58,60]. Autonomy is the need to feel that one’s behavior is autonomous and that the reasons for action are self-endorsed. The need for autonomy in educational settings is enhanced by choice, an explanation of the tasks, and the acceptance of one’s feelings, among others. The need for competence is defined as the need to feel effective and capable of performing various tasks or of experiencing effectiveness in one’s interaction with the world. Finally, relatedness means feelings of connectedness with significant others and a feeling of being accepted by other people. It is proposed that the frustration or satisfaction of these needs is a risk factor for a wide range of health-related problems and compensatory behaviors, including negative body image and eating disorders [60]. There is evidence that the satisfaction of the need for autonomy negatively predicts body image concerns in aerobics instructors [61]. According to the SDT, the extent to which BPNs are satisfied or thwarted depends on the social context [58,62]. Thus, sports environments that support mindfulness and positive embodiment should support the satisfaction of BPNs in exercisers. However, more research is needed on this topic.

There is evidence that mindfulness during physical activity is associated with higher internal exercise motivation [35,37], and that participation in mindful physical activities such as yoga increases intrinsic exercise motivation [36]. Mindfulness during physical activity is related to health and mood goals for exercise but not to appearance goals [35]. Specific pathways through which mindfulness might support autonomous physical activity motivation work through the satisfaction of BPNs. Individuals are naturally inclined to make choices that satisfy their innate needs (BPNs) to feel autonomous and competent. Non-judgmental and open awareness of physical body sensations might support feelings of competence and autonomy in physical activity (e.g., selecting a different physical activity or lowering the intensity of physical activity when feeling discomfort). This helps individuals to be more connected with themselves, to feel more satisfaction with physical activity, and to continue physical activity for long periods, gaining autonomy and competence over time.

Increased satisfaction with physical activity is one of the possible mechanisms through which the association between trait mindfulness and physical activity exists. Previous studies showed that trait mindfulness was related to physical activity through two consecutive mediators: first, state mindfulness (increased awareness) and then satisfaction [63,64]. Increased awareness helps people recognize positive changes, which contributes to their satisfaction with physical activity and helps to maintain it. According to the SDT, awareness of one’s internal experiences (such as thoughts, feelings, and sensations) and external conditions is key to supporting autonomous motivation [65]. However, more research is needed to understand the effect of different types of exercise on intrinsic exercise motivation, the satisfaction of BPNs, and pleasure during exercise as possible variables through which state body appreciation might be enhanced during exercise.

### 1.4. The Current Study

The main aim of the study was to examine the effects of an 8-week NF training program on state body appreciation and its correlates in female university-aged women in comparison with a control group undergoing functional training (FT). The immediate effect of exercise on positive body image, its correlates, and mindfulness was rarely tested in previous studies. We expected that NF training would have a significant positive effect on state body appreciation, state body surveillance and body functionality, perceived physical fitness, body–mind connection, mindfulness during physical activity, satisfaction of basic psychological needs, intrinsic motivation, and satisfaction during the exercise. We hypothesized that NF might be an important alternative to other approaches aiming to improve positive body image, such as participation in yoga and cognitive writing-based interventions. Previous studies reported that these approaches are effective for positive body image improvement [66,67]. In addition, we aimed to assess the mechanisms that might explain the changes in state body appreciation and other variables related to positive body image. Based on the DTE [26] and the Objectification Theory [68], we expected that the main changes in state body appreciation would be related to positive changes in mindfulness and body–mind connection during exercise and to a decrease in self-objectification.

## 2. Materials and Methods

### 2.1. Study Design and Ethical Considerations

We conducted a non-randomized, controlled trial involving two groups with pre- and post-test quantitative measures. Comparative measurements were performed in the experimental group (Nirvana Fitness, NF) and the control group (functional training, FT). This study was conducted in accordance with the Declaration of Helsinki and received ethics approval for the procedures at the Lithuanian Sports University with verification by the Committee for Social Sciences Research Ethics (Protocol No. SMTEK-131, 30 October 2022). All the participants provided written consent to participate in the study.

### 2.2. Participants and Procedure

Initially, through online registration during the autumn semester of 2023, a sample of 206 voluntarily registered female students from various universities/colleges in Lithuania was established. The final sample consisted of 43 students. The mean age of the study participants was 24.4 ± 6.7, range 19–36 years. All the study participants were studying at the bachelor level. When asked about their marital status, 30.2% reported being single, and the rest were in a committed relationship (married or living with a partner). Most of the students participating in the study were not employed (41.9%) or reported part-time employment (34.9%). Also, most of the study participants were in the range of healthy BMI (60.5%), while 27.9% were overweight/obese. The mean BMI was 23.4 ± 4.6 kg/m^2^, and the range of the BMI was 17.6–35.9 kg/m^2^.

The female volunteers were recruited online using multiple strategies. The recruitment entailed strategies such as posting information on social media network accounts (university/college platforms, the official platforms of student representatives, and the personal accounts of the investigators). Moreover, we invited females to participate in the intervention through oral announcements by the investigators in classes after obtaining permission from the administrative staff in the higher school institutions. Finally, the students we registered directed other students towards the intervention and motivated them to participate. The investigators involved in the study were responsible for recruiting the participants, and they developed an invitation flyer. The online flyer contained all the necessary information, contact information for the lead investigator, and an active link for registration. Interested students were included in the registration list via an active link with a request to complete an online questionnaire. The questionnaire was considered the first eligibility screening step. Recruitment continued for three weeks. Subsequently, using the online registration form, all the females were screened for the second eligibility step with an extra phone call. Additionally, those who met the eligibility criteria were contacted via email and provided with information regarding the scheduled first in-person meeting. After in-person contact, all the eligible and consenting (we requested written consent for participation) female (sex at birth) students were emailed with confirmation of their participation and information about the upcoming first exercise session. To select the sample, the following eligibility criteria were assigned: female gender (sex at birth) and written consent before participation. The exclusion criteria were as follows: a diagnosed eating disorder, an existing musculoskeletal or other chronic condition that could be considered a barrier to exercise, participation in professional sports, and attendance at Nirvana Fitness and/or FT in the past six months.

An initial sample size was calculated for an expectancy of a medium effect size (Cohen’s d = 0.5–0.6) with 80% statistical power and 5% at the statistical significance level, using two-sided statistical tests [69], and as a result, we aimed to recruit between 25 and 34 female students in each of the groups. The participants were assigned to the NF and FT groups. The allocation procedure for the groups was accomplished according to the individuals’ requests, considering their time commitments. In total, 50 participants agreed to be involved in the pre-test, while dropouts led to a total of 43 females being involved in the post-test. Of the participants who completed the pre- and post-tests, 21 belonged to the NF group, and 22 were from the FT group. The flowchart of the trial is shown in Figure 1.

The intervention was carried out during the autumn semester of the 2023/2024 university/college study year, with the first exercise session on 2nd October and the last on 30 November 2023. The first in-person introductory appointment with the participants was organized on 27 September 2023 in the Lithuanian Sports University auditorium, when the study was presented. The enrolled female students in the NF and FT groups took part in the exercise sessions twice a week for eight weeks. All the exercise sessions were held in the health and fitness club by agreement with the university.

After recruitment, eligibility evaluation, allocation, and group formation, both study groups completed assessments at two points in time. The enrolled participants completed the online survey, and the procedure for completing it had no time limit. The study procedures included pre- and post-test assessments. First, a pre-test assessment was conducted in both groups immediately after the first exercise session, using an emailed link to the questionnaire. After the first and last exercise session, the participants were requested to complete the questionnaire immediately after the workout session. The same self-reporting instruments were included in the pre-test and the post-test.

### 2.3. Intervention Description

A total of 16 h were allocated for the 8-week intervention, that is, participation twice a week for an hour’s exercise session for 8 weeks. The exercise sessions had different training content: NF training for the experimental condition and FT training for the control group. Professional instructors encouraged the participants in both training groups to attend the exercise sessions. Furthermore, twice a week, we emailed the students with a reminder about the upcoming exercise session to positively reinforce their attendance behavior. In addition, an online exercise session attendance form was created, which helped to register participation and retention rates. The participants of the intervention were instructed not to take part in any other fitness- and exercise-related physical activities during the eight-week period. The instructors who conducted the exercise sessions had more than ten years of health and fitness training experience and were qualified with a university degree in sport science. No specific training was provided for instructors. However, we asked them not to use appearance-based motivational language during exercise sessions. The content of the NF and FT sessions is described below.

The participants in the NF training condition attended two 60-min workouts per week. The body–mind were the key focuses of the workouts. This type of workout combines relaxation and stretching with yoga and Pilates movements. Combined with music, breathing, and flow in movement, exercisers can enter a state of relaxation, a flow state that fosters recharging, proper oxygenation, and a feeling of a refreshed mind. The directed 8-week training intervention covered four expressly defined choreographies, each of which consisted of eight sequences with four exercises each. Each workout included a warm-up with breathing activation and a cool-down with deep relaxation. The intensity of the NF workouts was low.

The participants in the FT group attended two 60-min FT workouts per week. Better performance of ordinary functions and daily tasks (tasks such as picking up a heavy object, walking up and down stairs, bending to pick something off the floor, playing with children, etc.) can be achieved through FT. FT is geared explicitly towards the use of the full body and the engagement of multiple muscles, especially the body core. The 8-week FT program incorporated different types of training with pre-determined content for each week as follows: stability and mobility training; body core training; weight training and training with light resistance; functional movement training; resistance circle training; resistance training; functional movement training; interval suspension training (total resistance exercise, TRX); and FT. Warm-ups and cool-downs were scheduled for each workout. The intensity of the workouts was moderate.

### 2.4. Measures

Demographic data: age, type of higher education institution (university/college), study cycle, marital status, and employment were the main features representing the demographic data of the female participants in the present study.

The State Body Appreciation Scale (SBAS-2) [70] was used to measure the extent to which the individuals appreciated and respected their bodies and had positive views of their bodies. The SBAS-2 is a modified version of the Body Appreciation Scale-2 (BAS-2) [71]. The SBAS-2 consists of nine statements. The participants were asked, “For each of the statements below, choose the option that best describes how you feel right now at this very moment." The answers are provided on a 5-point Likert-type scale ranging from 1 (strongly disagree) to 5 (strongly agree). Example items are “Right now, I respect my body” and “At this moment, I take a positive attitude toward my body." The total score was the mean of the scores for all the scale items. A higher score indicates greater state-of-body appreciation. The Lithuanian translation of the BAS-2 shows good psychometric properties [7]. The Lithuanian version of the SBAS-2 was used for the first time in the present study, and therefore we tested its psychometric properties. In this study, we also tested the factor structure of the SBAS-2 scale. The unidimensional model resulted in a good fit to the data with adequate model fit characteristics: CFI = 0.98, TLI = 0.97, RMSEA 0.076 (90% CI = 0.01–0.14), SRMR = 0.042. Internal consistency for the SBAS-2 was considered excellent: pre-test Cronbach’s α = 0.93 and post-test Cronbach’s α = 0.96.

The State Body Surveillance (SBS) subscale was used to assess state body surveillance during the exercise sessions. The Body Surveillance subscale of the Objectified Body Consciousness Scale (OBCS) [72] was modified to assess state body surveillance experience during physical activity rather than trait body surveillance. The SBS was measured using seven items from the 8-item Body Surveillance subscale, consistent with a previous study that employed the instrument to assess state rather than trait body surveillance during physical activity [37]. Sample items reflecting a measure of state body surveillance experience during exercise that has just been completed are as follows: “During the exercise, I thought about how I looked many times” and “During the exercise session, I was more concerned with how my body looks than what it can do." All items were measured on a 7-point Likert-type scale, from 1 (strongly disagree) to 7 (strongly agree). Higher values reflect greater state body surveillance. The Lithuanian version of the Body Surveillance subscale shows good psychometric properties [73]. In this study, parallel analysis indicated a one-factor solution, cumulatively accounting for 79.5% of the variance in the data. Also, internal consistency for the SBS was considered excellent: pre-test Cronbach’s α = 0.94 and post-test Cronbach’s α = 0.96.

The Functionality Appreciation Scale (FAS) was used to measure body functionality during physical activity. The original Functionality Appreciation Scale (FAS) [74] was revised to measure state-body functionality experience rather than trait-body functionality. The participants were asked to indicate their experience during the last exercise session. Example items are: “During the exercise, I appreciated my body for what it is capable of doing”; “During the exercise, I was grateful that my body enables me to engage in activities that I enjoy or find important”; and “During the exercise, I felt that my body does so much for me." The instrument consists of seven items assessed on a 5-point Likert-type scale, from 1 (strongly disagree) to 5 (strongly agree). A higher score reflects a greater state of appreciation for body functionality. The Lithuanian translation of the FAS and a determination that its psychometric properties are acceptable were presented in previous research [73]. In this study, the factor structure of the state FAS scale was additionally tested. A unidimensional model resulted in a good fit to the data, with model fit characteristics having acceptable values: CFI = 0.99, TLI = 0.97, RMSEA 0.077 (90% CI = 0.01–0.19), SRMR = 0.033. Internal consistency for the SFS was also considered excellent: pre-test Cronbach’s α = 0.90 and post-test Cronbach’s α = 0.92.

State Mindfulness in Physical Activity (SMS-PA-2) was used to assess state mindfulness during physical activity [37]. This 19-item self-reporting instrument distinguishes four subscales: 6-item Monitoring of the Mind, 6-item Monitoring of the Body, 3-item Accepting the Mind, and 4-item Accepting the Body. SMS-PA-2 includes statements such as “I felt present in my body” or “I was in tune with how hard my muscles were working." All items were measured on a 5-point Likert-type scale, ranging from 0 (not at all) to 4 (very much). The general score for each subscale was the mean of the scores for the items included in that subscale. The four general subscale scores were averaged to calculate the total SMS-PA-2 score, with a higher score indicating a greater state of mindfulness in physical activity. The Lithuanian translation of the SMS-PA-2 replicates the original four-factor structure, and a previous study has shown that it presents adequate psychometric properties [75]. Internal consistency of the measure was considered adequate to good: for Monitoring of the Mind, pre-test Cronbach’s α = 0.86 and post-test Cronbach’s α = 0.84; for Monitoring of the Body, pre-test Cronbach’s α = 0.74 and post-test Cronbach’s α = 0.83; for Accepting the Mind, pre-test Cronbach’s α = 0.63 and post-test Cronbach’s α = 0.64; and for Accepting the Body, pre-test Cronbach’s α = 0.77 and post-test Cronbach’s α = 0.91. For the general SMS-PA-2, pre-test Cronbach’s α = 0.81 and post-test Cronbach’s α = 0.89.

The Body–Mind Connection subscale (BMC) from the Physical Activity Body Experiences Questionnaire (PABEQ) [29] was used to assess body–mind connection during physical activity. The subscale items reflect the interaction between concepts such as thoughts, energy, physicality, awareness, and sense of self. Sample items are as follows: “I feel a connection between my physical energy level and the clarity of my thoughts” and “I have developed a connection between my body, my mind, and myself." All the items were measured on a 7-point Likert-type scale ranging from 1 (not at all true about me) to 7 (very true about me). The item responses were averaged to calculate the total score on the Body–Mind Connection subscale. Higher scores reflect stronger mind–body connections and indicate that the respondent was in a greater embodied state from the physical experience. Parallel analysis indicated a one-factor solution, cumulatively accounting for 70.1% of the variance in the data. Internal consistency for the BMC was good: pre-test Cronbach’s α = 0.93 and post-test Cronbach’s α = 0.89.

The Intrinsic Motivation subscale of the Behavioral Regulation in Exercise Questionnaire 2 (BREQ-2) was used to assess autonomous motivation during exercise [76]. BREQ-2 is a widely used, multidimensional, 19-item self-reporting instrument based on the Self-Determination Theory. It provides an assessment of five different types of autonomy-related behavioral regulation of physical activity and exercise: amotivation, external motivation, introjected motivation, identified motivation, and intrinsic motivation. In this study, only the 4-item Intrinsic Motivation subscale was used. The scale was revised to measure state experience with motivation. The participants were asked to indicate to what extent each of the following items was true about why she had engaged in the most recent exercise session. A sample item is as follows: “I felt pleasure and satisfaction while participating in this training." All items were measured on a 5-point Likert-type scale, ranging from 1 (not true) to 5 (very true). The responses to the four items were averaged to calculate the total state intrinsic motivation score, with a higher score indicating more state intrinsic regulation of physical activity and exercise. The Lithuanian translation of the BREQ-2 shows good psychometric properties [77]. Internal consistency for the State Intrinsic Motivation subscale was good: pre-test Cronbach’s α = 0.89 and post-test Cronbach’s α = 0.92.

The Basic Psychological Needs Satisfaction in Physical Education Scale (BPNSS) [78] was used to assess the satisfaction of basic psychological needs during physical activity. The 12-item self-reporting measure includes three subscales and provides an assessment of autonomy (a 4-item autonomy subscale), competence (a 4-item competence subscale), and relatedness (a 4-item relatedness subscale). This instrument was modified to assess state experience during the last exercise session rather than trait satisfaction of basic psychological needs during exercise. To assess state experience, the first question was modified from “in general in PE” to “during the first (last) exercise session." Also, the words “physical education” and “lessons” were changed to “exercise sessions” or "workouts," and “children” and "classmates" were changed to “members of the exercise group” or “other exercisers." Examples of the statements are as follows: “I felt that the exercise session was provided in the way I would like it to be." (autonomy); “I felt that I improved even if the tasks were considered difficult by most of the other exercisers” (competence); and “My relationships with members of the exercise group were very friendly” (relationships). The participants were asked to provide their responses on a 7-point Likert scale ranging from 1 (I do not agree at all) to 7 (I completely agree), with a midpoint of 4 (I moderately agree). A higher score reflects greater satisfaction of the autonomy, competence, and relatedness needs during physical activity. The Lithuanian version of the scale had never been tested before, and therefore we ran exploratory (EFA) and confirmatory (CFA) factor analyses. The Kaiser–Meyer–Olkin (KMO) test resulted in a measure of sampling adequacy of 0.82, and Bartlett’s test of sphericity (χ2 = 362.6, df = 66, *p* < 0.001) indicated that it was appropriate to proceed with the exploratory factor analysis. Parallel analysis indicated that three factors should be retained, cumulatively accounting for 75.7% of the variance in the data. The principal component analysis revealed that all the items were correlated to the factors of their original subscales. The first factor was composed of the items from the competence subscale and explained 26.7% of the variance; the second incorporated items from the relatedness subscale and provided 25.0% of the data variance; and the last combined the autonomy subscale items and explained 24.0% of the variance. Finally, the original three-factor structure was confirmed with satisfactory model fit indices: CFI = 0.96, TLI = 0.94, RMSEA 0.090 (90% CI = 0.01–0.15), and SRMR = 0.065. The internal consistency of the measure was considered adequate: for autonomy, pre-test Cronbach’s α = 0.79 and post-test Cronbach’s α = 0.91; for competence, pre-test Cronbach’s α = 0.83 and post-test Cronbach’s α = 0.90; and for relatedness, pre-test Cronbach’s α = 0.79 and post-test Cronbach’s α = 0.86.

The Empirical Valence Scale (EVS) [79] was used to assess satisfaction and pleasure during exercise. A single item was applied: “Using the scale below, please mark the ONE number that best reflects the overall amount of pleasantness or unpleasantness that you felt during the workout you have just done. The response scale ranges from −10 (very unpleasant workout) to +10 (very pleasant workout), with 0 indicating a neutral position.”

Perceived physical fitness (PPF) was tested by the self-developed question, “What do you think your fitness level is? How do you feel among your peers when it comes to climbing stairs or a hill, running, or doing physical work?” The answer options were as follows: 1 (too weak), 2 (a little weak), 3 (my fitness level is the same as most of my peers), 4 (a little stronger than others), and 5 (much stronger than others). Higher values correspond to greater PPF. This question has been used in previous studies [80].

### 2.5. Statistical Analysis

First, the descriptive statistics were used to test the distribution normality and possible outliers of the continuous variables. The internal consistency of the study measures was tested by Cronbach’s α at two points, at the pre-test and post-test. A Cronbach’s α over 0.65 was considered adequate [81], although it should generally be noted that Cronbach’s α values are sensitive to the number of items included in the scale [82]. Next, the construct validity of the study measures was studied by performing exploratory factor analysis (EFA) and then confirmatory factor analysis (CFA). The cut-off values for each model fit index were used as recommended by Hu and Bentler: RMSEA ≤ 0.06 for good fit and ≤0.08 for acceptable fit; SRMR ≤ 0.08 for good fit and ≤0.12 for acceptable fit; CFI ≥ 0.95 for good fit and ≥ 0.90 for acceptable fit [83].

Further, to compare the state experience during the first exercise session between the NF and FT groups, the study variables were compared using the independent sample Student’s *t*-test. Next, to compare the pre-test and post-test results, paired-sample t-test statistics were used. The data are presented as means and SDs alongside Cohen’s d effect sizes, which were classified as small (0.2–0.4), medium (0.5–0.6), and large (≥ 0.8) [84]. In addition, to calculate the effects of time and the interaction between time and the group on the study measures, a repeated measures ANOVA was employed. The partial eta-squared (ŋ^2^) represents effect sizes: 0.01 and below 0.06 were considered small, 0.06 and below 0.14 as medium, and ≥ 0.14 as large [84]. Last, to predict the change in state body appreciation during the intervention, a three-step hierarchical linear regression analysis was conducted, and the change in model summary statistics (∆R2, ∆F, and p of ∆F) was presented after each step. The significance threshold was set at a *p*-value of < 0.05. Statistical analyses were conducted using IBM SPSS Statistics 29 (IBM Corp., Armonk, NY, USA) and Mplus v8.7 (Muthén & Muthén, Los Angeles, CA, USA).

## 3. Results

The mean age of the study participants in the NF and FT groups was as follows: 25.3 ± 9.2 and 23.4 ± 2.8 years (*p* = 0.259). The range of the BMI was 17.6–35.9 kg/m^2^ and did not differ significantly between groups (NF = 23.9 ± 4.7 kg/m^2^ and FT = 22.9 ± 4.6 kg/m^2^, *p* = 0.430).

Most of the study participants (67.4%) reported no lifetime involvement in any sports and/or physical activity. Looking at the duration of the previous engagement in sport, there were no significant differences between the NF and FT groups (4.8 ± 4.0 and 7.1 ± 6.3 years, respectively, *p* = 0.245). One study participant in the FT group reported previous participation in the same exercise sessions, and three had previously participated in NF exercise sessions (two in the FT group and one in the NF group). The mean number of exercise sessions attended during the intervention did not differ significantly between the NF and FT groups: NF 11.2 ± 3.1 and FT 11.0 ± 4.0 (*p* = 0.845).

Further, Table 1 represents the comparison of the state experience during the first exercise session between the NF and FT groups. No significant differences across the groups were observed, except that the Body–Mind Connection subscale score was initially higher in the NF group.

The changes in state experiences when comparing the first exercise session versus the last are provided in Table 2. In the NF group, the Monitoring of the Mind and Monitoring of the Body subscale scores from the SMS-PA-2 increased during the exercise, as did the total score. Notably, state functionality appreciation, body appreciation, and satisfaction with basic psychological needs improved significantly during the intervention, while the body surveillance score decreased. Also, an increase in the Body–Mind Connection score was observed. In parallel, similar changes were observed in the FT group except as regards the improvement in satisfaction during the exercise of the need for autonomy and relatedness. Also, a small but significant increase in perceived physical fitness scores was observed in the FT group. No significant change in the state intrinsic motivation scores during the exercise was observed in either of the groups. All changes demonstrated medium-to-high effect sizes. In addition, most of the study measures demonstrated significant time effects, except for the SMS-PA-2: Accepting Body subscale, intrinsic motivation, and satisfaction during the exercise, whereas a time x group interaction effect was observed only for state improvement in relatedness, underlying the higher effect in the NF group.

In the last step, hierarchical linear regression for predicting the change in state body appreciation was implemented (Table 3). In Step 1, the change in body–mind connection and state mindfulness significantly predicted the change in body appreciation during the intervention, and the model was significant (*p* = 0.002). However, in Step 2, where changes in state body surveillance, state functionality appreciation, and perceived physical fitness were added, we found that only state mindfulness and body surveillance significantly predicted change in body appreciation, and the R2 change from Step 1 (R2 = 0.32) to Step 2 (R2 = 0.55) was significant (*p =* 0.002). In Step 2, the overall model was also significant (*p* < 0.001). In Step 3, the motivation-related study measures were added: change in satisfaction and pleasure during exercise, change in state intrinsic exercise motivation, and change in the subscales of the state Basic Psychological Needs Satisfaction Scale: competence, relatedness, and autonomy. It was revealed that only positive changes in state mindfulness, body surveillance, and relatedness significantly predicted improvement in state body appreciation during the exercise intervention. The R2 change from Step 2 (R2 = 0.55) to Step 3 (R2 = 0.66) was not significant (*p* = 0.104). Notably, the type of exercise (NF vs. FT) did not demonstrate any effect on the change in state body appreciation. However, the final model explained 66% (*p* < 0.001) of the state-body appreciation during the exercise.

## 4. Discussion

### 4.1. The Main Findings

The main aim of this non-randomized controlled intervention was to investigate whether NF, a mindfulness-based physical activity, might be a practice that increases positive body image (body appreciation) in university-aged women. In the present study, we hypothesized that NF might be an important alternative to other approaches for improving positive body image, such as yoga and cognitive writing-based interventions, whose effectiveness for the enhancement of positive body image has been established [66,67]. In addition, we assessed whether the practice of NF would lead to improvements in body–mind connection (embodiment) and mindfulness during the physical activity, a decrease in state body surveillance, and an improvement in functionality appreciation. We tested the mechanisms that might explain the changes in state body appreciation and other variables related to positive body image. In line with our hypothesis, women who participated in and completed an 8-week NF exercise program reported improvements in state body appreciation, body–mind connection, mindfulness during exercise (increased monitoring of mind and body), body surveillance, satisfaction during exercise, and satisfaction of basic psychological needs, with medium to high effect sizes. No changes were observed in intrinsic motivation or perceived physical fitness. Interestingly, almost the same significant improvements in the analyzed variables were observed in the control group that participated in FT. However, in the FT group, no improvements were observed in satisfaction during physical activity or satisfaction of the psychological needs of autonomy and relatedness. Our further analysis of the results showed that the potential mechanisms through which the NF program improved state body appreciation were related to enhancement of mind–body connection, increased state mindfulness during physical activity, decreased body surveillance, and satisfaction of the basic psychological need for relatedness.

### 4.2. Participation in Nirvana Fitness as a Mindfulness-Based Physical Activity and Its Effect on State Body Appreciation and Its Correlates

Our findings are in line with the literature analyzing the associations between mindfulness-based physical activity and body appreciation [10,11,37,70]. The novelty of the present study is that the state version of body appreciation and other body image-related constructs, including mindfulness, were tested during physical activity. A small number of studies tested the effect of prolonged physical activity on the immediate experience of body appreciation and other variables related to positive body image. Understanding the experiences and changes related to body image during physical activity is important for sport science research since it helps to give an understanding of the importance of the content and nature of an exercise program on the immediate body image and motivation experience of women. Understanding these experiences might inform future physical activity and positive body image promotion programs for women and girls.

The novel finding of the present study is that NF, one of the modern types of mindfulness-based physical activity programs, improves the positive experience of body appreciation during exercise in women, and the effect of exercise works through an increase in embodiment and mindfulness during physical activity, lower body surveillance, and greater satisfaction of the psychological need for relatedness. Previous experimental studies have also reported that mindfulness-based physical activity (yoga) positively affects body appreciation through enhanced body–mind connection and mindfulness [19,35,42] and decreased body surveillance [19,35,42,49]. In the present study, we tested the immediate effect of physical activity on state body image and made a comparison between the first and last exercise sessions in the 8-week period. A previous study that tested state mindfulness during yoga reported no significant changes after participation in a 16-week program, while an increase in trait mindfulness was associated with a change in trait body appreciation after 16 weeks of yoga participation [85]. Since there is a lack of studies assessing changes in state body appreciation as an outcome of various types of exercise, it is important to continue this type of research.

In modern fitness, yoga is departing from its spiritual foundations and is increasingly being assimilated into an exercise and fitness culture that is driven by appearance and commercial considerations [86]. Scholars have discussed how the Sanskrit nomenclature of ancient yoga has been replaced by terminology that is easier to understand, emphasizing the physiological benefits of yoga practice and including a variety of asana routines to prevent boredom among modern exercisers [87]. NF is a modern configuration of yoga-based movement that was created to meet the expectations of health and fitness exercisers. However, like modern yoga, NF focuses the attention of the exercisers on the mindful monitoring of body movement and breathing and the acceptance of various thoughts and feelings of the body, including negative ones. Therefore, mindfulness during this type of exercise increases, and exercisers shift their attention towards the body’s functionality rather than its appearance. Our results support the DTE, which holds that the fundamental element of positive embodiment is a deep and authentic connection between body–mind and the ability to hear and prioritize the voice of the body over external requirements and social demands [22,26,28]. Our results also support the objectification theory, which states that a less objectifying environment and situation might reduce body surveillance [88,89]. This is also in line with the concept of being attuned to exercise, according to which mindful exercise is any movement done with compassion and attention to oneself, self-acceptance, and joy. Mindful exercise is process-oriented, aiming for wellness and inner harmony, whereas unconscious exercise is often focused on appearance and/or athletic or appearance-related results [40].

In the present study, we also observed positive changes in state body appreciation and its correlates in the control group that exercised FT. These findings suggest that FT, as a physical activity that traditionally is not assumed to be mindfulness-based, might also positively affect state body appreciation and its correlates, such as state body surveillance, functionality appreciation, and body–mind connection. FT is a different exercise type than Nirvana Fitness, with a higher exercise intensity [90]. FT is a strong worldwide fitness trend [91]. FT involves the integrated and balanced development of a wide range of physical abilities to ensure independence, efficiency, and safety in daily, occupational, or sporting activities. FT uses strength exercises emphasizing the development of stability in the center of the body, characterized by integrated, multi-joint or multi-segment, asymmetrical, multi-plane, acyclic, speedy, and unstable movements [92]. FT involves movements that fully engage the attention of the exerciser and focus it on the functionality of the body rather than on its appearance. In our study, we also observed that positive changes in perceived physical fitness occurred in the FT group but not in the NF group, suggesting that the higher-intensity exercise had a greater effect on perceived physical fitness compared with the lower-intensity NF exercise. Thus, positive changes in state body evaluation and its correlates in the FT group may be related to positive perceptual changes in physical fitness. However, in the present study, we did not test the mechanisms through which the positive changes in FT participants occur. Nevertheless, it seems that both types of exercise might be beneficial for the improvement of state body appreciation and its correlates. FT and its effects on state body appreciation should be further investigated in future studies.

### 4.3. Mindfulness, Changes in Internal Motivation, Satisfaction of Basic Psychological Needs, and Satisfaction during Exercise

In the present study, we observed no changes in internal motivation during the NF program, but the satisfaction of basic psychological needs was significantly improved. These results are in line with studies that showed that yoga positively increases the satisfaction of basic psychological needs in students [36]. It seems that mindfulness during physical activity and becoming more aware of mental and physical sensations during physical activity might help exercisers experience greater satisfaction with their needs for autonomy, competence, and relatedness. The results of the hierarchical linear regression showed that mindfulness during exercise, body surveillance, and satisfaction of the psychological need for relatedness significantly predicted state body appreciation. These findings partially confirm the acceptance model of body appreciation and intuitive eating [93]. According to this model, unconditional general acceptance is related to body acceptance by others, and body acceptance by others is associated with body appreciation through body surveillance. Our finding suggests that when exercisers become more mindful during exercise, they concentrate on their immediate body–mind experiences and possibly invest less effort in observing and judging the appearance of other exercisers; they may therefore feel more safe about their own body appearance, which in turn decreases body surveillance. This process possibly helps to satisfy the psychological need for relatedness during exercise, and relatedness positively affects body appreciation. However, this finding is new, and future studies should continue to investigate the satisfaction of basic psychological needs during exercise and the related changes in state body appreciation.

SDT proposes that the extent to which basic psychological needs are satisfied depends on the social context [60]. A basic process model of self-determination theory in health and physical education contexts has been developed [62,94]. According to this model, support for basic psychological needs in a social context is related to the level of satisfaction of basic psychological needs at the intrapersonal level, and the satisfaction of needs further leads to the development of internal motivation during physical activity. Studies show that internal motivation is associated with various positive outcomes, including higher leisure-time exercise motivation, higher satisfaction during physical activity, and more positive self-esteem and body image [59,94,95,96,97]. Based on this model of motivational sequence, the results of the present study suggest that in a mindfulness-based physical activity environment, basic psychological needs are well supported, and the increase in relatedness positively affects the body appreciation of exercising women as an outcome of physical activity. However, in the present study, we cannot understand if increased autonomy, competence, and relatedness further increase internal exercise motivation or if the increased satisfaction of BPNs is an outcome of the increased satisfaction during physical activity that was also observed in the present study. Therefore, more findings on this topic are needed to test this theoretical model on exercising adults, and we recommend that future studies continue this type of research.

### 4.4. The Effect of Mindfulness-Based Physical Activity on Perceived Physical Fitness

An important finding was that perceived physical fitness did not significantly change in the NF group. NF workouts are of low to moderate intensity, and evidence exists that the impact of low-intensity physical activity on body satisfaction is lower than the impact of physical activity of higher intensity [16,18]. It was previously reported that perceived physical fitness is associated with higher body satisfaction [14], but the results of the present study show that increased mindfulness and attentiveness to the body–mind might have a positive impact on state body appreciation even in a situation in which perceived physical fitness remains unchanged. In other words, low-intensity physical activity might have an effective impact on body appreciation during physical activity if this activity is mindful, and exercising women might experience state body appreciation during exercise even if they do not actually perceive increases in physical fitness. However, future studies should test these findings.

### 4.5. Limitations and Future Directions

The main limitation of this non-randomized controlled intervention is that the participants were not randomly allocated to the experimental and control conditions. The exercise sessions were organized at around midday and overlapped with lectures and other responsibilities of the students; therefore, the attrition numbers were high, and we could not implement randomization. It is impossible to answer the question of whether the same pattern of findings would have emerged if randomization had been implemented. Fortunately, the groups were homogenous and similar in the measured variables. In future studies, it will be important to organize exercise sessions by providing two different times for participating students so that they can adjust according to what is possible for them. The relatively small sample size prevented us from doing deeper statistical analyses of the results. Thus, the generalizability of the results is limited.

Another important limitation of the present study is that the type of exercise in the control group was very different from the type of exercise in the experimental group. To compare the state of body appreciation and its correlates, as well as the state of mindfulness during physical activity, we needed to compare two exercising groups. However, the type of exercise was very different in our experimental and control groups. In future studies, we recommend comparing groups with similar exercise types and intensities (e.g., NF and yoga, FT and running or Cross-Fit, etc.).

A final limitation of the present study is the lack of objectively assessed data such as BMI and physical activity during exercise and leisure time. This is an important area for future research. Obtaining these data would help to better control the bias of the study. Therefore, it is recommended that future studies do this. The main strength of the present study is that it is based on a sound theoretical background. Also, in the present research, we not only assessed changes in mindfulness, body image, and motivation-related variables but also tested the mechanisms through which these changes occur. A previous systematic review concluded that intervention studies focusing on the links between body image and movement in girls and women favor theoretical justification and explanation of the mechanisms of change [17].

It is recommended that future studies be carried out to assess other types of physical activity (with different physical activity intensities) and test the effects of mindfulness on the state of body appreciation in women. The positive changes in the control group who took part in a FT program in the present study suggest that physical activity of a higher intensity, such as FT, might also be beneficial for positive body image improvements in women. In our previous study, we observed that mindfulness during physical activity mediates the associations between participation in various types of physical activity and body appreciation in physically active students [38]. Thus, it is important to test the role of mindfulness in future studies in which other types of physical activity, such as FT, are tested. The participation of more diverse samples in interventions is also recommended, including women with a body mass index higher than the healthy range, the elderly, and people with different sexual identities. Follow-up testing after this program is also recommended. Future studies that try to replicate our findings would be valuable in indicating whether the effects of the present study are reliable.

## 5. Conclusions

In line with the DTE, Objectification Theory, and SDT, an 8-week Nirvana Fitness program was found to have a positive impact on state body appreciation, state body surveillance and functionality appreciation during exercise, body–mind connection, state mindfulness, general exercise satisfaction, and satisfaction of basic psychological needs during exercise in university-aged women. The main predictors for the change in state body appreciation were positive changes in body–mind connection, state mindfulness, body surveillance, and satisfaction of the psychological need for relatedness. The functional training program also positively impacted state body image and its correlates and mindfulness during exercise, suggesting that both exercise types might be beneficial for state body image improvement. These findings highlight the importance of targeting mindfulness, body surveillance, and relatedness during physical activity in university-aged women, as these are factors that can contribute to improvements in positive body image as an immediate effect of exercise.

## Figures and Tables

**Figure 1 healthcare-12-01632-f001:**
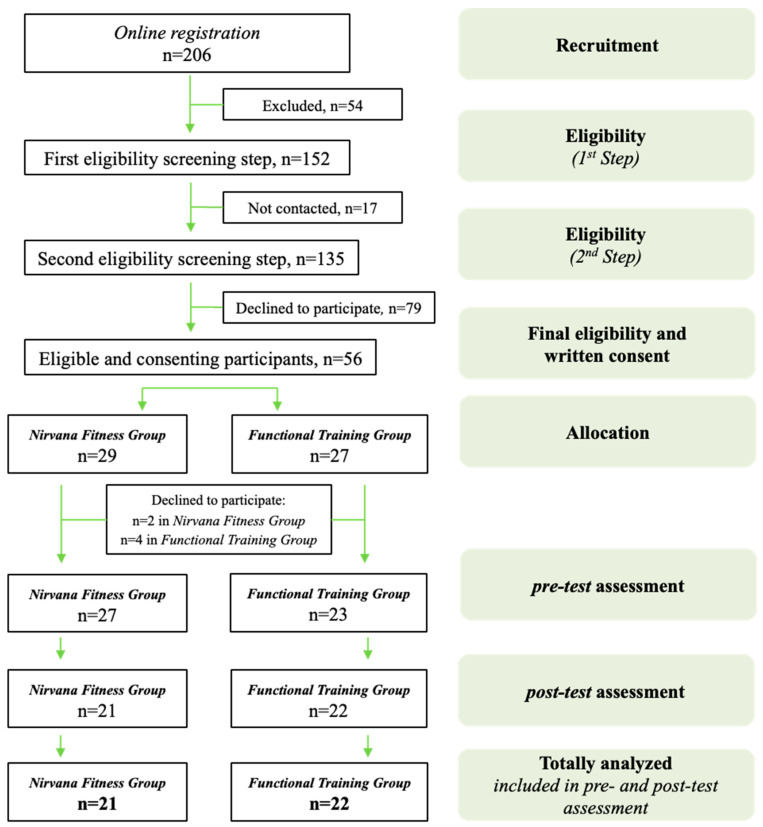
Flowchart describing participant enrolment and flow throughout the trial.

**Table 1 healthcare-12-01632-t001:** Comparison of the state measures, mind–body connection, and perceived physical fitness scores during the first exercise session between Nirvana Fitness and the functional training groups (n = 43).

Variables	Nirvana Fitness Group (n = 21)	Functional Training Group (n = 22)	Cohen’s d	*p*
SMS-PA-2: MM	3.17 ± 0.79	2.82 ± 0.87	-	0.175
SMS-PA-2: MB	3.35 ± 0.56	3.49 ± 0.70	-	0.465
SMS-PA-2: AM	3.02 ± 0.81	2.96 ± 0.99	-	0.803
SMS-PA-2: AB	3.56 ± 0.75	3.39 ± 0.79	-	0.467
SMS-PA-2: total score	3.30 ± 0.51	3.18 ± 0.50	-	0.462
BREQ-2: Intrinsic Motivation	4.66 ± 0.50	4.28 ± 0.87	-	0.133
Satisfaction during exercise	8.29 ± 1.85	6.86 ± 3.09	-	0.077
State Body Appreciation Scale 2	3.89 ± 0.69	3.46 ± 0.75	-	0.056
OBCS: Body Surveillance	2.88 ± 1.27	3.70 ± 1.91	-	0.105
Functionality Appreciation Scale	4.12 ± 0.54	3.77 ± 0.70	-	0.076
BPNSS: Competence	4.63 ± 0.86	4.59 ± 1.47	-	0.913
BPNSS: Relatedness	3.62 ± 1.19	3.42 ± 1.19	-	0.594
BPNSS: Autonomy	4.82 ± 1.00	4.51 ± 1.26	-	0.378
PABEQ: Mind–Body Connection	4.66 ± 1.34	3.40 ± 1.10	1.03	0.002
Perceived physical fitness	2.90 ± 1.09	2.55 ± 1.10	-	0.324

Note. SMS-PA-2—State Mindfulness Scale for Physical Activity 2, MM—Monitoring Mind, MB—Monitoring Body, AM—Accepting Mind, AB—Accepting Body, BREQ-2—Behavioral Regulation in Exercise Questionnaire 2, OBCS—Objectified Body Consciousness Scale, BPNSS—Basic Psychological Need Satisfaction Scale, PABEQ—Physical Activity Body Experiences Questionnaire.

**Table 2 healthcare-12-01632-t002:** Comparison of the state study variables scores, body–mind connection and perceived physical fitness at the baseline and after the intervention within the groups, time and interaction between the group and time effects (n = 43).

Variables	Nirvana Fitness Group (n = 21)		Cohen’s d	*p*	Functional Training Group (n = 22)		Cohen’s d	*p*	The Effect of Time, ŋ^2^; *p*	The Interaction Effect between the Intervention Group and Time, ŋ^2^; *p*
	Pre	Post			Pre	Post				
SMS-PA-2: MM	3.17 ± 0.79	3.72 ± 0.67	0.93	<0.001	2.82 ± 0.87	3.60 ± 0.74	0.76	0.002	0.40; <0.001	0.02; 0.388
SMS-PA-2: MB	3.35 ± 0.56	4.12 ± 0.60	1.18	<0.001	3.49 ± 0.70	4.08 ± 0.68	1.13	<0.001	0.58; <0.001	0.02; 0.324
SMS-PA-2: AM	3.02 ± 0.81	3.41 ± 0.97	-	0.130	2.96 ± 0.99	3.50 ± 0.67	0.48	0.035	0.15; 0.01	0.006; 0.633
SMS-PA-2: AB	3.56 ± 0.75	3.73 ± 0.84	-	0.332	3.39 ± 0.79	3.61 ± 0.82	-	0.079	0.08; 0.063	0.002; 0.771
SMS-PA-2: total score	3.30 ± 0.51	3.82 ± 0.59	1.12	<0.001	3.18 ± 0.50	3.75 ± 0.51	1.08	<0.001	0.56; <0.001	0.002; 0.758
BREQ-2: Intrinsic Regulation	4.66 ± 0.50	4.76 ± 0.38	-	0.397	4.28 ± 0.87	4.25 ± 0.92	-	0.873	0.002; 0.769	0.008; 0.570
Satisfaction during exercise	8.29 ± 1.85	9.29 ± 0.90	0.52	0.023	6.86 ± 3.09	7.05 ± 3.93	-	0.281	0.03; 0.238	0.02; 0.412
SBAS-2	3.89 ± 0.69	4.35 ± 0.63	0.64	0.008	3.46 ± 0.75	3.93 ± 0.86	0.77	0.002	0.34; <0.001	0.00; 0.959
OBCS: SBS	2.88 ± 1.27	2.12 ± 0.87	0.68	0.008	3.70 ± 1.91	2.57 ± 1.68	0.79	0.004	0.36; <0.001	0.02; 0.349
FAS	4.12 ± 0.54	4.59 ± 0.44	1.11	<0.001	3.77 ± 0.70	4.28 ± 0.62	0.91	0.001	0.50; <0.001	0.001; 0.808
BPNSS: Competence	4.63 ± 0.86	5.82 ± 0.81	1.31	<0.001	4.59 ± 1.47	5.17 ± 1.37	0.53	0.016	0.45; <0.001	0.09; 0.055
BPNSS: Relatedness	3.62 ± 1.19	4.87 ± 1.28	1.06	<0.001	3.42 ± 1.19	3.76 ± 1.36	-	0.102	0.37; <0.001	0.17; 0.006
BPNSS: Autonomy	4.82 ± 1.00	5.43 ± 1.34	0.66	0.009	4.51 ± 1.26	4.84 ± 1.52	-	0.183	0.13; 0.016	0.01; 0.463
PABEQ: Mind–Body Connection	4.66 ± 1.34	5.36 ± 1.03	0.66	0.007	3.40 ± 1.10	4.36 ± 1.24	1.21	<0.001	0.45; <0.001	0.02; 0.388
Perceived physical fitness	2.90 ± 1.09	3.10 ± 0.89	-	0.248	2.55 ± 1.10	3.09 ± 1.07	0.45	0.046	0.12; 0.023	0.03; 0.261

Note. ŋ^2^—partial eta-squared, SMS-PA-2—State Mindfulness Scale for Physical Activity 2, MM—Monitoring Mind, MB—Monitoring Body, AM—Accepting Mind, AB—Accepting Body, BREQ-2—Behavioral Regulation in Exercise Questionnaire 2, OBCS—Objectified Body Consciousness Scale, SBS—State Body Surveillance Scale; FAS—Functionality Appreciation Scale; SBAS-2—State Body Appreciation Scale 2, BPNSS—Basic Psychological Need Satisfaction Scale, PABEQ—Physical Activity Body Experience Questionnaire.

**Table 3 healthcare-12-01632-t003:** Hierarchical linear regression predicting change in state body appreciation during the intervention (n = 43).

Study Variables	B	β	*t*	*p*	VIF
Step 1					
NF group vs. FT group	0.08	0.06	0.47	0.642	1.02
∆ Mind–body connection from PABEQ	0.27	0.38	2.82	0.008	1.02
∆ State mindfulness from SMS-PA-2	0.56	0.41	3.07	0.004	1.01
Model summary: R^2^, F, p	0.32, 5.99, 0.002				
Step 2					
NF group vs. FT group	0.16	0.12	1.04	0.306	1.06
∆ Mind–body connection from PABEQ	0.15	0.20	1.67	0.105	1.18
∆ State mindfulness from SMS-PA-2	0.48	0.35	2.52	0.016	1.51
∆ State body surveillance from OBC	0.25	0.48	3.63	<0.001	1.37
∆ Functionality appreciation	0.08	0.06	0.36	0.718	1.85
∆ Perceived physical fitness	0.04	0.07	0.55	0.586	1.13
Model summary: R^2^, F, p	0.55, 7.20, <0.001				
Model summary change: ∆R^2^, ∆F, p ∆F	0.23, 6.07, 0.002				
Step 3					
NF group vs. FT group	−0.14	−0.10	−0.78	0.439	1.56
∆ Mind–body connection from PABEQ	0.02	0.03	0.20	0.847	2.02
∆ State mindfulness from SMS-PA-2	0.67	0.48	3.28	0.003	1.93
∆ State body surveillance from OBC	0.30	0.59	4.44	<0.001	1.58
∆ Functionality appreciation	0.11	0.08	0.52	0.605	2.13
∆ Perceived physical fitness	0.03	0.04	0.38	0.707	1.22
∆ Satisfaction and pleasure during exercise	0.004	0.02	0.12	0.906	2.47
∆ State intrinsic exercise regulation from BREQ-2	−0.12	−0.14	−0.82	0.420	2.64
∆ State competence from BPNSS	0.17	0.26	1.81	0.081	1.88
∆ State relatedness from BPNSS	0.23	0.39	2.41	0.022	2.36
∆ State autonomy from BPNSS	−0.10	−0.19	−1.16	0.257	2.38
Model summary: R^2^, F, p	0.66, 5.40, <0.001				
Model summary change: ∆R^2^, ∆F, p ∆F	0.11, 2.02, 0.104				

Note. B—unstandardized, β—standardized regression coefficient, *t*—*t*-test of B, VIF—variance inflation factor, NF—Nirvana Fitness, FT—functional training, ∆—change during the intervention, PABEQ—Physical Activity Body Experience Questionnaire, SMS-PA-2—State Mindfulness Scale for Physical Activity 2, OBC—Objectified Body Consciousness, BREQ-2—Behavioural Regulation in Exercise Questionnaire 2, BPNSS—Basic Psychological Need Scale.

## Data Availability

The dataset available on request from the authors.

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
