# Peer review of "Evaluating the Impact of the Nirvana Fitness and Functional Training Programs on Young Women’s State Body Appreciation and Its Correlates"

_healthcare, 2024, doi:10.3390/healthcare12161632_

Round 1
Reviewer 1 Report
Comments and Suggestions for Authors
The work seems well conducted and structured.
The theoretical constructs are sufficiently analyzed.
The study design is clear. The selection procedures are sufficiently clear. The statistical analysis is appropriate.
However, the interventions (NT and FT) needs to be described more specifically in terms of duration, type of exercises, and intensity.
If possible, I would include the questionnaires and scales used as attachments or supplementary materials.
The main limitation of the study is the lack of objective data regarding health-related parameters and daily or weekly physical activity -unstructured activity- (example: How much physical activity was performed compared to what is recommended by international guidelines?)
Is it possible for the authors to trace some of these values? We might have results related only to perception, but without a real preventive benefit.
I think it is essential to explore, in discussion section, the importance of perceived physical fitness and its relationship with subjective correlates.
It does not seem clear in the discussion and conclusion whether NT intervention is preferable to FT. Could the authors explain this concept better?

Author Response
Reviewer 1
Thank you for your time reviewing our paper and for your comments. All changes made in the text are highlighted in blue font.
The work seems well conducted and structured.
The theoretical constructs are sufficiently analyzed.
The study design is clear. The selection procedures are sufficiently clear. The statistical analysis is appropriate.
Thank you.
However, the interventions (NT and FT) needs to be described more specifically in terms of duration, type of exercises, and intensity.
Thank you for this comment. The duration and type of exercises were described in the first version of the manuscript. However, we have now included information about the intensity of the workouts.
If possible, I would include the questionnaires and scales used as attachments or supplementary materials.
The questionnaires and scales used are well-known and widely used instruments that might be found elsewhere. Moreover, we used national language-validated versions of instruments that we think are not relevant to international readers.
The main limitation of the study is the lack of objective data regarding health-related parameters and daily or weekly physical activity -unstructured activity- (example: How much physical activity was performed compared to what is recommended by international guidelines?)
Is it possible for the authors to trace some of these values? We might have results related only to perception, but without a real preventive benefit.
Thank you for this important comment. We included it as the limitation of the study and developed recommendations for future studies.
I think it is essential to explore, in the discussion section, the importance of perceived physical fitness and its relationship with subjective correlates.
We assessed the importance of perceived physical fitness for state body appreciation (see Table 3). The results clearly show that the perception of physical fitness is not a significant predictor of state body appreciation change in the NF group. The perceived physical fitness may have more significance for positive body appreciation changes in the FT group, but this was not the aim of the study to assess it. However, we expanded the discussion on this topic in the Discussion.
It does not seem clear in the discussion and conclusion whether NT intervention is preferable to FT. Could the authors explain this concept better?
Thank you very much for this important comment. We expanded the discussion about it in the Discussion and Conclusions.

Reviewer 2 Report
Comments and Suggestions for Authors
The introduction part is very well grounded from a theoretical point of view and all the basic elements that are present in the study are adequately covered. The aim of the paper is clearly formulated: the main aim of the study was to examine the effects of an 8-week NF training program on body condition assessment and its correlates in young women, compared to a control group undergoing functional training (FT). But we have some problems with the originality part of the present article. There are far too many similarities with two other similar articles namely: https://doi.org/10.3389/fpsyg.2024.1412259 and https://doi.org/10.3389/fspor.2024.1360145. The recommendation is that in the first step these similarities should be resolved, according to the ratio of similarity coefficients I have attached.
1. Previous studies have also suggested that mindfulness during physical activity is related to health and mood goals for exercise, but is not related to appearance goals [35]. The recommendation is to cite all the studies you mention or rephrase the sentence.
2. We hypothesized that NF might be an important alternative to other approaches aiming to improve positive body image, such as participation in yoga and cognitive writing-based interventions, whose effectiveness for the enhancement of positive body image has been established [66,67]. If we are discussing a hypothesis of the study, it should not contain citations, and the recommendation is that it should be included in the Materials and Methods chapter.
3. The novelty of the present study is that the state version of body appreciation and other body image-related constructs, including mindfulness, were tested during physical activity. The part with the novel elements of the study, which is under discussion: it is recommended to be shifted to the introduction part, after the aim of the paper.
4. Previous studies have shown that perceived physical fitness is associated with higher body satisfaction [14],. Please kindly rephrase or cite all previous studies.
5. In our previous study, we observed that mindfulness during physical activity mediates the associations between participation in various types of physical activity and body appreciation in physically active students [38]. I have noticed that in the limitations of the study there is an article with a lot of similarities, but this aspect with the two articles needs to be rectified.

Minor editing of the English language is needed.
Author Response
Reviewer 2
Thank you for your time reviewing our paper and for your comments. All changes made in the text are highlighted in blue font.
The introduction part is very well grounded from a theoretical point of view and all the basic elements that are present in the study are adequately covered. The aim of the paper is clearly formulated: the main aim of the study was to examine the effects of an 8-week NF training program on body condition assessment and its correlates in young women, compared to a control group undergoing functional training (FT).
Thank you.
But we have some problems with the originality part of the present article. There are far too many similarities with two other similar articles namely: https://doi.org/10.3389/fpsyg.2024.1412259 and https://doi.org/10.3389/fspor.2024.1360145. The recommendation is that in the first step these similarities should be resolved, according to the ratio of similarity coefficients I have attached.
- Previous studies have also suggested that mindfulness during physical activity is related to health and mood goals for exercise, but is not related to appearance goals [35]. The recommendation is to cite all the studies you mention or rephrase the sentence.
Thank you, we rephrased the sentence.
- We hypothesized that NF might be an important alternative to other approaches aiming to improve positive body image, such as participation in yoga and cognitive writing-based interventions, whose effectiveness for the enhancement of positive body image has been established [66,67]. If we are discussing a hypothesis of the study, it should not contain citations, and the recommendation is that it should be included in the Materials and Methods chapter.
Thank you, we revised the hypothesis to avoid citations. Nevertheless, we think that the hypothesis should be included at the end of the Introduction, but not in the Methods section.
- The novelty of the present study is that the state version of body appreciation and other body image-related constructs, including mindfulness, were tested during physical activity. The part with the novel elements of the study, which is under discussion: it is recommended to be shifted to the introduction part, after the aim of the paper.
Thank you for this comment. We added a sentence emphasizing the novelty of the study right after the aim of the study.
- Previous studies have shown that perceived physical fitness is associated with higher body satisfaction [14],. Please kindly rephrase or cite all previous studies.
Thank you, rephrased.
- In our previous study, we observed that mindfulness during physical activity mediates the associations between participation in various types of physical activity and body appreciation in physically active students [38]. I have noticed that in the limitations of the study there is an article with a lot of similarities, but this aspect with the two articles needs to be rectified.
These are two different design studies. We double-checked the similarities between the ref. 38 and our manuscript limitation sections and concluded that all coincidences in the text are random and related to common structures of sentences describing limitations in all articles.

Reviewer 3 Report
Comments and Suggestions for Authors
Thank you for the opportunity to review your study. The manuscript is in general well-structured and addresses an interesting topic. I have several suggestions that I hope you find useful:
Line 3. Please consider using "female college students" instead of "young women" to be more specific, as all participants were college students (Lines 240-242).
Line 243. Did authors examine age as a potential confounding factor? Given the age range of 19-36 years old, this may impact study results, as the outcome variables could potentially tie to age.
Line 269. Please specify whether this refers to sex at birth or self-reported gender identity.
Line 309. Did the instructors receive training for this study? How did authors ensure consistency among instructors? Was any fidelity check conducted? This is a critical issue that needs to be clearly addressed.
Line 313. How did authors control potential influences such as college PE courses, club activities, and other group or individual physical activities on both groups?
Line 348. Please provide more details on how authors "tested its psychometric properties" and include relevant results. Unlike SBAS-2 and FAS, authors did not provide factor analysis results for SBS, SMS-PA-2, PABEQ, and BREQ-2. Was this omission intentional, and if so, is there a specific reason?
Line 494. Given the nature of the design and potential covariates, ANCOVA might be more appropriate than ANOVA.
Table 2. The title row could be improved, especially the last two column titles, which could be reorganized to be more concise/straightforward.
In addition, this study had a relatively small sample size. This limitation should be discussed in terms of its impact on statistical power and the generalizability of the results.
Author Response
Reviewer 3
Thank you for your time reviewing our paper and for your comments. All changes made in the text are highlighted in blue font.
Thank you for the opportunity to review your study. The manuscript is in general well-structured and addresses an interesting topic. I have several suggestions that I hope you find useful:
Line 3. Please consider using "female college students" instead of "young women" to be more specific, as all participants were college students (Lines 240-242).
Thank you, corrected.
Line 243. Did authors examine age as a potential confounding factor? Given the age range of 19-36 years old, this may impact study results, as the outcome variables could potentially tie to age.
Thank you for this important comment. The study sample was designed to be homogenous according to age: 88,4% of the university-aged women in both groups were up to 28 years old and only a few (5 women) were older. Moreover, the results of a recent study showed that body appreciation (the outcome of the study) is not related to age https://doi.org/10.1371/journal.pone.0306913 Also, all the participants were healthy female students studying at several universities with the majority in the range of normal body mass.
Line 269. Please specify whether this refers to sex at birth or self-reported gender identity.
Thank you, we specified in the text that sex at birth was assessed.
Line 309. Did the instructors receive training for this study? How did authors ensure consistency among instructors?
No training for instructors was provided. They were only asked not to use appearance–based motivational language during the workouts. We included this information in the manuscript.
Was any fidelity check conducted? This is a critical issue that needs to be clearly addressed.
A fidelity check was conducted by filling out the online exercise session attendance form. This information was included in the first version of the manuscript.
Line 313. How did authors control potential influences such as college PE courses, club activities, and other group or individual physical activities on both groups?
The participants of the intervention were instructed not to take part in any other fitness and exercise-related physical activities during the eight-week period. We now included this explanation in the manuscript.
Line 348. Please provide more details on how authors "tested its psychometric properties" and include relevant results. Unlike SBAS-2 and FAS, authors did not provide factor analysis results for SBS, SMS-PA-2, PABEQ, and BREQ-2. Was this omission intentional, and if so, is there a specific reason?
The psychometric properties of SBS, SMS-PA-2, PABEQ, and BREQ-2 instruments were tested in previous our studies in students‘ samples (national language translated instruments reliability (i.e. internal consistency, parallel forms of reliability), validity (content, construct, face and criterion validity) and referencing for these articles were presented in the first version of the manuscript.
Line 494. Given the nature of the design and potential covariates, ANCOVA might be more appropriate than ANOVA.
As explained before, we decided not to include covariates in the analysis due to the homogeneous sample.
Table 2. The title row could be improved, especially the last two column titles, which could be reorganized to be more concise/straightforward.
Thank you, the title row in Table 2 was corrected to be more explanatory.
In addition, this study had a relatively small sample size. This limitation should be discussed in terms of its impact on statistical power and the generalizability of the results.
Thank you for this comment. We now discussed it as the limitation.

Round 2
Reviewer 2 Report
Comments and Suggestions for Authors
The introduction part is very well grounded from a theoretical point of view and all the basic elements that are present in the study are adequately covered. The aim of the paper is clearly formulated: the main aim of the study was to examine the effects of an 8-week NF training program on body condition assessment and its correlates in young women, compared to a control group undergoing functional training (FT). But we have some problems with the originality part of the present article. There are far too many similarities with two other similar articles namely: https://doi.org/10.3389/fpsyg.2024.1412259 and https://doi.org/10.3389/fspor.2024.1360145. The recommendation is that in the first step these similarities should be resolved, according to the ratio of similarity coefficients I have attached.

Moderate editing of English language required.
Author Response
The introduction part is very well grounded from a theoretical point of view and all the basic elements that are present in the study are adequately covered. The aim of the paper is clearly formulated: the main aim of the study was to examine the effects of an 8-week NF training program on body condition assessment and its correlates in young women, compared to a control group undergoing functional training (FT). But we have some problems with the originality part of the present article. There are far too many similarities with two other similar articles namely: https://doi.org/10.3389/fpsyg.2024.1412259 and https://doi.org/10.3389/fspor.2024.1360145. The recommendation is that in the first step these similarities should be resolved, according to the ratio of similarity coefficients I have attached.
Thank you for drawing our attention to the similarities between the manuscript and two of our previous papers. We have revised the submitted paper again and note that the overlap with our two previously published papers is within acceptable limits. Similarities are atypical, related to terms and concepts used in other scientific works and our previously published works written based on the same scientific project. We would also like to draw your attention to the fact that when checking for plagiarism, the system should disable the check for similarities in references.
